# Slow recombination of spontaneously dissociated organic fluorophore excitons

Takahiko Yamanaka[1,2]*, Hajime Nakanotani[2,3]* & Chihaya Adachi [2,3]*

The harvesting of excitons as luminescence by organic fluorophores forms the basis of light-emitting applications. Although high photoluminescence quantum yield is essential for efficient light emission, concentration-dependent quenching of the emissive exciton is generally observed. Here we demonstrate generation and accumulation of concentration-dependent "long-lived" (*i.e.*, over 1 h) photo-generated carriers and the successive release of their energy as electroluminescence in a solid-state film containing a polar fluorophore. While fluorophore excitons are generally believed to be stable because of their high exciton binding energies, our observations show that some of the excitons undergo spontaneous exciton dissociation in a solid-state film by spontaneous orientation polarization even without an external electric field. These results lead to the reconsideration of the meaning of "*luminescence quantum yield*" for the solid films containing polar organic molecules because it can differ for optical and electrical excitation.

[1] Central Research Laboratory, Hamamatsu Photonics K.K., 5000 Hirakuchi, Hamakita-ku, Hamamatsu, Shizuoka 434-8601, Japan. [2] Center for Organic Photonics and Electronics Research (OPERA), Kyushu University, 744 Motooka, Nishi-ku, Fukuoka 819-0395, Japan. [3] International Institute for Carbon-Neutral Energy Research (WPI-I2CNER), Kyushu University, 744 Motooka, Nishi-ku, Fukuoka 819-0395, Japan. *email: takahiko.yamanaka@crl.hpk.co.jp; nakanotani@cstf.kyushu-u.ac.jp; adachi@cstf.kyushu-u.ac.jp

O rganic fluorophores dispersed in a host matrix form the basis of a variety of molecular optoelectronic systems for light-emitting applications, including light-emitting diodes[1], bio-imaging techniques[2], photon up-conversion systems[3], luminescent solar concentrators[4], and long-persistent luminescent materials[5]. In all of these applications, the choice of a material system with a high photoluminescence quantum yield (PLQY) is essential for achieving highly efficient emission. However, the PLQY of fluorophores in a host matrix is generally sensitive to the doping concentration, limit the fluorophores that can be used.

In this study, we mainly focused on the excited-state dynamics of a highly-doped polar organic fluorophore exhibiting thermally activated delayed fluorescence (TADF)[6] in host molecules, because TADF has attracted great interest for organic light-emitting applications, such as organic light-emitting diodes (OLEDs)[7] and long-persistent luminescence (LPL)[5], and has been widely investigated over the last decades[8,9]. The interpretation that some of the excitons in the fluorophores undergo non-radiative relaxation to the ground state through a non-adiabatic interaction between two potential surfaces, i.e., internal conversion[10], or the deactivation via dipole–dipole interaction between neighbor fluorophores[11] is widely accepted. However, fluorophores exhibiting TADF are generally intramolecular donor–acceptor (D–A) systems and naturally form a charge-transfer (CT) excited state in both the lowest singlet and lowest triplet excited states ($S_1$ and $T_1$). Thus, another nonradiative decay process should be considered: exciton dissociation, i.e., carrier generation, through the CT-type exciton as an intermediate state which has been widely recognized in photovoltaic devices containing a mixture of donor and acceptor molecules[12]. While the concentration-dependent quenching of the emissive exciton, generally called concentration quenching, is well known and usually attributed to dipole–dipole interactions, we unveil here an additional, non-negligible concentration quenching mechanism based on spontaneous excitons dissociation that has so far been overlooked.

## Results

### Photo-physical properties of TPA-DCPP.
To find evidence of spontaneous exciton dissociation in organic fluorophore alone in a solid matrix, we first evaluated the fundamental photoluminescence (PL) properties of a model fluorophore, namely, TPA-DCPP[13] (Fig. 1a) because it possesses a large permanent dipole moment (dipole moment = 13.05 D). The PLQY of TPA-DCPP doped in a CBP host matrix at different concentrations (the absorption and emission spectra are shown in Supplementary Fig. 1a, b) decreased with an increase of the doping concentration as shown in Fig. 1a, indicating the concentration quenching processes such as the deactivation of exciton via dipole–dipole interaction and the internal conversion process coexist in the highly doped films. Here note that the exciton quenching due to a back energy transfer from $T_1$ of TPA-DCPP (2.25 eV) to $T_1$ of CBP (2.55 eV) is negligible because triplet energy level of CBP matrix is higher than that of TPA-DCPP. Although the time constant of the delayed fluorescence component ($\tau_d$: the radiative decay constant of the singlet excitons that are generated via reverse intersystem crossing from $T_1$ of TPA-DCPP) was estimated to be 50 μs in a 10-wt%-TPA-DCPP:CBP film, $\tau_d$ becomes shorter with an increase of doping concentration (Fig. 1b). This trend indicates that the deactivation rate of triplet excitons on TPA-DCPP increases with a reduction of the average distance between TPA-DCPP molecules.

To unravel the dynamics of the quenching process of the triplet excitons, the dependence of the time-resolved PL (TRPL) decay

on the photo-excitation pulse width was measured in a 50-wt%-TPA-DCPP:CBP film (Fig. 1c). A long PL decay component with a millisecond-order decay lifetime, which is longer than $\tau_d$, was observed only when the film was excited with long pulses (>10 μs). This long-lived PL does not originate from phosphorescence, because the long-lived emission lifetime is independent of the detection wavelength from 650 nm to 800 nm (Supplementary Fig. 2). These observations indicate that, following long-pulse excitation, a long-lived emission component that is different from "normal" triplet exciton decay, i.e., TADF, occurs in the blend film even after the excitation light is turned off when the film was excited by the long-excitation pulse width. Further, since the emission intensity in a 50-wt%-TPA-DCPP:CBP film shows the almost linear response for the excitation light power (Supplementary Fig. 3), the effect of the bimolecular exciton annihilation process can be excluded. As the long PL decay component follows a power-law dependence (Supplementary Fig. 4), the origin of the long-lived emission can be considered as the recombination of separated carriers[14,15] generated by exciton dissociation.

### External-electric-field-modulated TRPL.
Since the motion of the carriers can be modulated by an electric field, we prepared a non-carrier-injection-type device to study the effect of an external electric field on the TRPL decay. A hole-blocking layer (T2T) and electron-blocking layer (CBP) were placed adjacent to the 50-wt%-TPA-DCPP:CBP layer (emission layer; EML) to prevent carrier injection from each electrode into the EML (Fig. 2a, and Supplementary Fig. 5). Additionally, the T2T and CBP layers contain almost no absorbance at the excitation wavelength (470 nm), meaning that only TPA-DCPP is directly excited (Supplementary Fig. 6). Figure 2b shows the results of the electric-field-modulated luminescence measurement. Even in the device architecture equipped with electrodes, long-lived emission similar to that in the case without electrodes was observed. When a forward-bias voltage pulse was applied to the ITO at one millisecond after turning off the excitation light, an emission spike was clearly observed. Since the emission spike is not observed under the same conditions without prior photo-excitation (Supplementary Fig. 7), the emission spike can be attributed to carriers accumulated inside the EML and not injected carriers.

The dependence of the TRPL profile on the timing of the external voltage pulse was also measured in the same device. As shown in Fig. 2c, an emission spike appeared in the TRPL profile upon application of a voltage pulse regardless of the delay time. The TRPL tail intensity after applying the external electric field becomes appreciably weaker compared to the intensity when an electric field was not applied, again indicating that carriers generated by photo-excitation are being consumed by recombination. More interestingly, the emission spike intensity is relatively constant for delays ranging over 1 ms despite the gradual decrease in the long emission tail, and the spike could still be detected even after 1 h (Supplementary Fig. 8a, b), indicating the existence of ultra-long-lived photo-generated carriers in the EML.

The origin of the long-lived photo-generated carriers leading to luminescence upon application of an external electric field can be ascribed to exciton dissociation by an internal electric field. This hypothesis is reinforced by the following experimental results. First, the TRPL decay and spike intensity are affected by applying a reverse bias during and after excitation (Supplementary Fig. 9a). The TRPL decay appreciably reduced for a reverse bias of −10 V compared to with no biasing. In the case of a low bias voltage (−10 V), since the potential slope in the EML is rather steep, the photo-generated carriers are rapidly separated and then kept apart under the influence of the field, so the carriers recombine

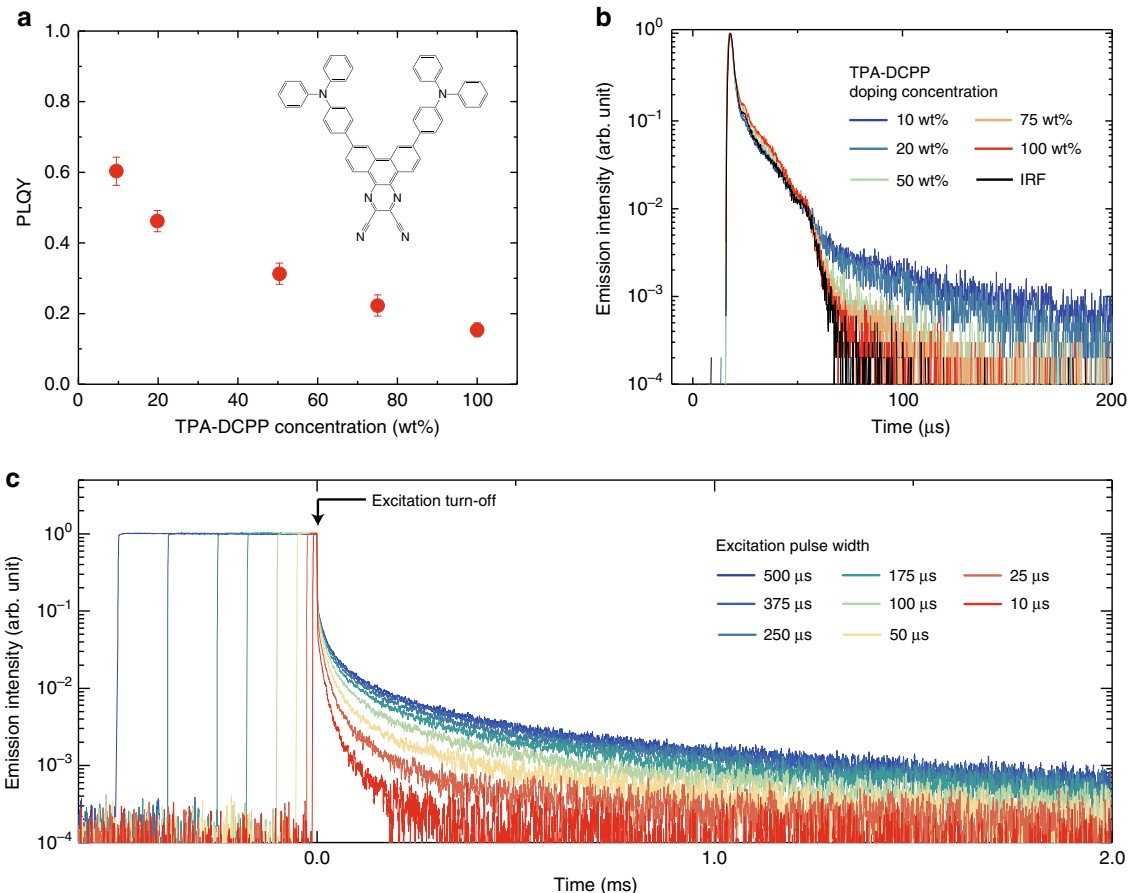

**Fig. 1 Fundamental PL properties of TPA-DCPP-based solid-state films. a** Concentration dependence of PLQY in TPA-DCPP:CBP blends. The error bars are estimated by taking into account of instrument accuracy and experimental variability. Inset: chemical structure of TPA-DCPP. **b** TRPL profiles in TPA-DCPP:CBP blends with different TPA-DCPP doping concentration. The black line indicates the instrument response function (IRF) of the fluorescence lifetime spectrometer used for the measurements. **c** Dependence of TRPL profile on excitation pulse width for a 50-wt%-TPA-DCPP:CBP blend. A 470-nm light source was used as excitation light, and the power density was fixed at 25 mW cm$^{-2}$. The lifetime of the long-tailed component saturated for pulses longer than width of 375 μs.

inefficiently. Second, when multiple voltage pulses are applied to the sample, the spike emission intensity decreases with each pulse (Supplementary Fig. 9b), indicating the stepwise consumption of carriers. Finally, the TRPL decay diminishes with an increase of excitation power density (Supplementary Fig. 9c). This can be understood as a shortening of the diffusion length of the carriers because of an increase of recombination probability associated with the denser triplet generation. This result also excludes the possibility of carrier generation through two-photon ionization of TPA-DCPP[15].

**The role of SOP in exciton dissociation event**. To gain more insight into the mechanism, we conducted displacement current measurements (DCM)[16] of an OLED with a TPA-DCPP:CBP layer as the EML. As shown in Fig. 3a, we found a clear threshold-voltage shift depending on the doping concentration, indicating a larger accumulation of interfacial charges with an increasing concentration of TPA-DCPP (from 20 wt% to 50 wt%). Furthermore, DCM was also performed on the device used for the TRPL measurements to verify the accumulated carriers (Fig. 3b). In the first scan after light irradiation accompanied with a bias of −10 V, the transient current peak appeared at 2.5 V. The current peak, however, disappeared in the second scan. This is again consistent with accumulated carriers being consumed as recombination current.

Based on the DCM results, we can explain the driving force for carrier generation in the solid films. Since a permanent dipole moment of CBP is close to zero and TPA-DCPP has a large permanent dipole moment (13.05 D), the threshold-voltage shift observed in the DCM (Fig. 3a) can be attributed to the formation of a concentration-dependent potential slope across the EML by spontaneous orientation polarization (SOP)[17,18] of TPA-DCPP molecules. Since the concentration-dependent potential slope means a change of polarity of the EML with increasing the doping concentration, the PL spectrum of the highly doped codeposited film shows a redshifted spectrum as shown in Supplementary Fig. 1b. In other words, the observation of the redshifted spectrum with increasing the doping concentration is not conflicting with our proposed mechanism. In fact, the intensity of the emission spike in a 20-wt%-doped non-injecting device is much weaker than that of a 50-wt%-doped one (Supplementary Fig. 9d), which is due to a lower exciton dissociation probability. Since the accumulated interfacial charge density at the 50-wt %-doped EML interface is calculated to be −1.6 mC m$^{-2}$, the large surface potential slope can be 54.9 mV nm$^{-1}$ at least. The potential slope should be formed across the solid film, so some of the photo-generated excitons on TPA-DCPP can spontaneously dissociate even without an external applied electric field (Fig. 3c). The origin of the long-lived emission observed in the solid film thus can be attributed to the recombination of holes and electrons that were generated by spontaneous exciton dissociation. We

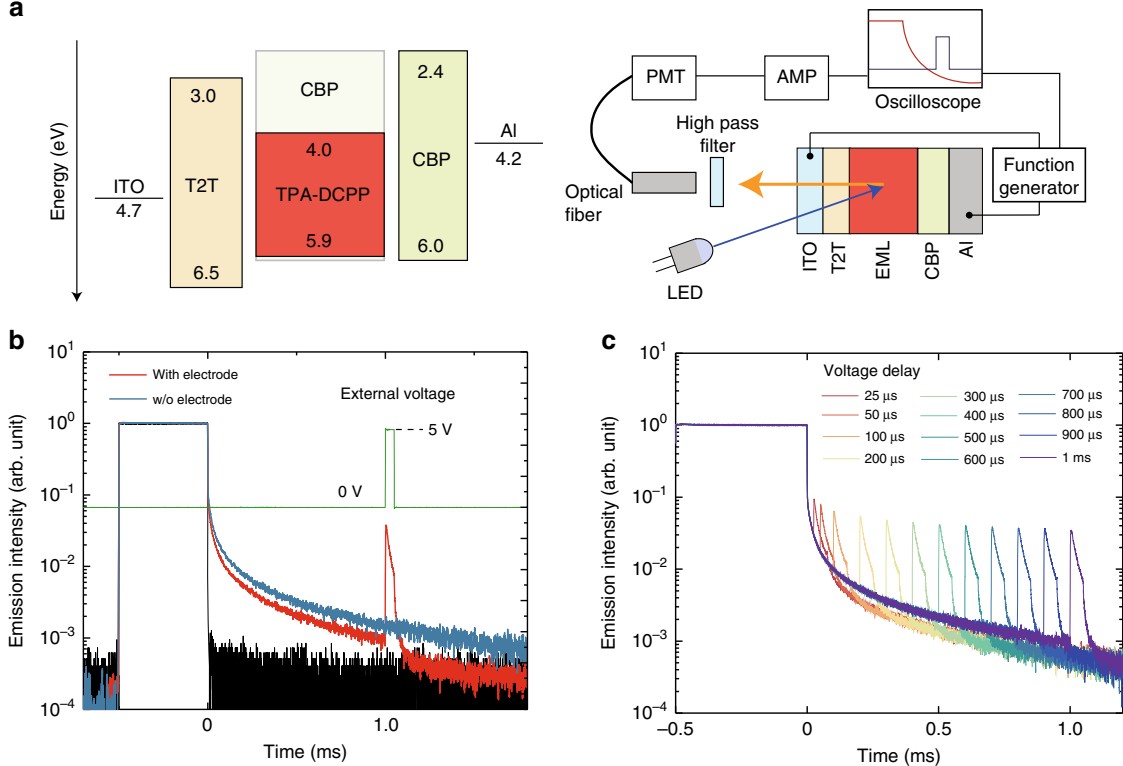

**Fig. 2 Device structure, experimental set-up, and external-electric-field-modulated luminescence. a** Energy-level diagram of materials used in this work (left) and schematic of the experimental set-up for external-electric-field-modulated luminescence measurements (right). ITO and Al electrode were used as anode and cathode, respectively. **b** External-electric-field-modulated TRPL profile in a device with a 50-wt%-TPA-DCPP:CBP blend (red line). The blue line is the TRPL profile of a 50-wt%-TPA-DCPP:CBP blend (without electrodes or T2T and CBP layers) for comparison. The black and green line indicates the IRF in this measurement system and the time profile of the external applied voltage, respectively. The excitation light pulse width was set to 500 μs. **c** Dependence of the external-electric-field-modulated TRPL profiles on external-voltage-pulse delay in the 50 wt%-TPA-DCPP:CBP device. The applied voltage and the width were fixed to 5 V and 50 μs, respectively. Note that it is not a multiple voltage pulses. This experiment used independent voltage pulses for each timing, and multiple measurements were performed.

roughly calculate the yield of long-lived emission to total emission is ~1%, so the dissociation yield should be much higher than this. Compared with the total exciton quenching yield, the dissociation yield is actually quite small, but the fact that the process exists cannot be ignored. Here, we discuss the role of spin states in the spontaneous exciton dissociation by the SOP. The triplet excitons of polar molecules, i.e., CT-type molecule exhibiting TADF, are considered to dissociate because of (1) the long exciton lifetime, and (2) the small energy splitting between singlet CT and triplet CT energy levels. Further, the disappearance of the delayed fluorescence component with increasing the doping concentration suggests the quenching of the triplet exciton. However, it cannot be determined which spin state, singlet and triplet, is responsible for the exciton dissociation at this stage. While we need to study it in more detail, it is evident that excitons are spontaneously dissociated by SOP.

## Discussion
Although long-lived emission was completely absent from a film doped with the non-polar fluorophore TBRb, long-lived emission was observed from a film doped with the polar fluorophore 4CzIPN[6] (3.85 D), which also exhibits SOP[19] (Fig. 3d). Further, even when the voltage offset was set to 0 V, the non-injection-type devices based on 4CzIPN:CBP blends show clear EL spikes when an external voltage pulse is applied. This result indicates that the excitons of 4CzIPN are spontaneously dissociated and diffused by the SOP. Also, of course, when the voltage offset was set to −10 V on the TBRb-based device, a clear response of the EL

spike due to the external voltage pulse was observed. This result shows that the excitons of TBRb dissociate by the "external" electric field. However, when the voltage offset was set to 0 V, no response was observed because of a lacking of SOP. Therefore, our proposed mechanism can explain the additional concentration-dependent quenching process in polar fluorophores. Furthermore, as the larger spatial separation[20] of the hole and electron in intermolecular D–A systems leads to a much weaker exciton binding energy compared to that of intramolecular D–A systems, the driving force for charge separation in exciplex-based LPL systems[5] containing 2,8-Bis(diphenyl-phosphoryl)-dibenzo[b,d]thiophene, which has a moderate dipole moment (4.60 D), can also be well explained by our proposed mechanism. Additionally, the long-lived decay disappears completely in transient electroluminescence profiles (Supplementary Fig. 10) of OLEDs because the applied voltage promotes carrier recombination rather than charge separation. This means that there is an additional exciton quenching process under optical excitation that does not exist under electrical excitation condition. Thus, the luminescence quantum yield of solid films containing polar fluorophores or polar host matrix can differ for optical and electrical excitation, especially in the case of exciplexes[21].

## Methods
**Materials**. The molecules 4′-bis(carbazol-9-yl) biphenyl (CBP), 7,10-bis(4-(diphenylamino)phenyl)-2,3-dicyanopyrazinophenanthrene (TPA-DCPP), 2,4,6-tris(biphenyl-3-yl)-1,3,5- triazine (T2T), and 2,7-bis(2,2′-bipyridine-5-yl)triphenylene (BPy-TP2) were purchased from NARD institute, Ltd. The molecules 1,1-bis[(di-4-tolylamino)phenyl]cyclohexane (TAPC) and 2,8-di-tert-butyl-5,11-

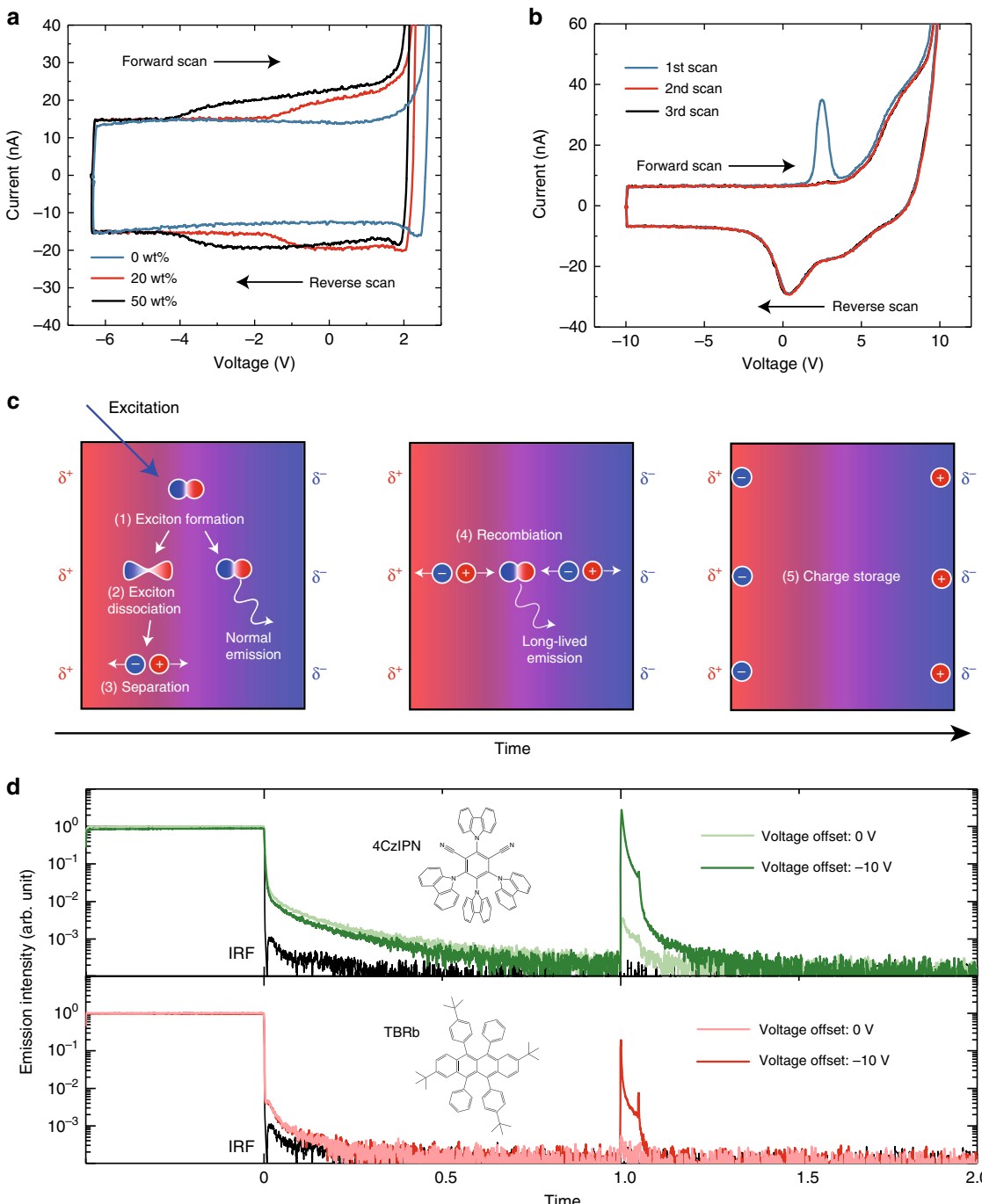

**Fig. 3 Displacement current measurements (DCM) and the observation of long-lived emission from other fluorophores. a** DCM profiles in OLEDs with EMLs of TPA-DCPP:CBP having different TPA-DCPP doping concentrations. The device structure was ITO (100 nm)/TAPC (50 nm)/EML (30 nm)/T2T (20 nm)/BPy-TP2 (60 nm)/LiF (1.6 nm)/Al (100 nm). **b** Dependence of DCM profiles on the number of scans in non-injecting TPA-DCPP:CBP devices with the structure in Fig. 2a. **c** Schematic of spontaneous exciton dissociation and recombination processes in solid films containing organic polar fluorophores. **d** TRPL profiles in non-injecting device based on 50-wt%-TBRb:CBP and 50-wt%-4CzIPN:CBP blend as EML with the structure in Fig. 2a. The black line indicate IRFs. A long PL decay component having a millisecond-order decay lifetime that is much longer than the intrinsic time constant of TADF ($\tau_d = 4.6 \, \mu s$) was observed in the 50-wt%-4CzIPN:CBP blend. In these measurements, a UV-LED with a peak wavelength of 340 nm was used as the excitation light source. Note that the ITO electrode slightly absorbs the excitation light and emits light. The applied peak voltage and the width were fixed to 5 V and 50 μs, respectively.

bis(4-tert-butylphenyl)-6,12-diphenyltetracene (TBRb) were purchased from Luminescence Technology Corp. The molecule 2,4,5,6-tetra(9H-carbazol-9-yl)iso-phthalonitrile (4CzIPN) was synthesized according to the literature (7).

**Sample preparation and measurement procedures of photoluminescence properties**. All organic layers were formed by thermal evaporation under high-

vacuum conditions ($<10^{-4}$ Pa). Organic films with a thickness of 50 nm were grown on precleaned quartz substrates. The non-injecting devices were fabricated on clean tin-doped indium oxide (ITO) glass substrates and had an effective device area of 4 mm$^2$. The device structure for external-electric-field-modulated luminescence measurements was ITO (100 nm)/T2T (30 nm)/EML (100 nm)/CBP (30 nm)/aluminum (Al) (100 nm), with ITO as the anode, T2T as the hole-blocking layer, CBP and TPA-DCPP co-deposited as the emissive layer, neat CBP as the

electron-blocking layer, and Al as the cathode. After device fabrication, the devices were immediately encapsulated with glass lids and epoxy glue in a dry nitrogen-filled glove box. PLQY of the samples and the PL spectrum were measured using an absolute photoluminescence quantum yield measurement system (C11347-01, Hamamatsu Photonics). Absorption spectrum of the samples was recorded on ultraviolet-visible spectrometer (Lambda 950-PKA, PerkinElmer).

**External-electric-field-modulated luminescence characterization**. TRPL decay was measured using a fluorescence lifetime spectrometer (C11367, Hamamatsu Photonics) with a time-correlated single photon counting (TCSPC) method for measuring the time constant of TADF molecules. We also used a photo-sensor module (H10721-01, Hamamatsu Photonics), an amplifier unit (C11184, Hamamatsu Photonics), and an oscilloscope (WaveRunner 640Zi, Teledyne Lecroy) with a function generator (WaveStation2012, Teledyne Lecroy) as an external electric field source for observing the long-lived TRPL and external-electric-field-modulated TRPL profile measurements. A pulsed LED driving circuit was prepared as an excitation source to control the excitation pulse width, and a long-pass filter with an optical density of 6 was used for extraction of the sample signal. All TRPL signals measured with the oscilloscope were averaged 10,000 times and normalized at the intensity just before turning off the photo-excitation.

**DCM characterization**. DCM was conducted using a current input preamplifier (LI-76, NF Corporation) and multifunction filter (3611, NF Corporation) with the oscilloscope and the function generator as noted earlier. All DCM curves were measured with a sweep rate of $10\,V\,s^{-1}$ at room temperature.

## Data availability
The data that support the findings of this study are available from the corresponding authors upon reasonable request.

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

## Acknowledgements
We acknowledge the helpful discussions with Shigeo Hara and Toru Hirohata. We also acknowledge W. J. Potscavage Jr. for his assistance with preparation of this paper.

## Author contributions
The project was conceived and designed by T.Y. and H.N. T.Y. prepared the samples and conducted all measurements. T.Y. and H.N. analyzed all of the data. C.A. supervised the project. All authors contributed to writing the paper and critically commented on the project.

## Competing interests
The authors declare no competing interests.
