## [Peer Review File · Nature Communications]

Reviewers' comments:

Reviewer #1 (Remarks to the Author):

In this manuscript, Yamanaka et al studied the long tail of fluorescence emissions and assigned its origination to the spontaneous exciton dissociation in the solid state. Though they pointed some interesting results, some statements are not so accurate. For materials with CT states, it has been widely recognized that charge recombination and dissociation exist simultaneously, which has been studied in photovoltaic systems. Therefore, for TADF materials, such processes would coexist and such behaviors can vary with different molecules and different dopant concentrations. Though the author demonstrated the dissociation process of TADF emitters, they did not provide more information about how to control it to enhance the performance of TADF emitters in devices. Therefore, I think major revisions are required:

- (1) In the manuscript, the author found that "the TRPL decay diminishes with an increase of excitation power density. This can be understood as a shortening of the diffusion length of the carriers because of an increase of recombination probability associated with the denser triplet generation" and "the intensity of the 106 emission spike in a 20-wt%-doped non-injecting device is much weaker than that of a 50-wt%-doped 107 one (Extended Data Fig. 3d), which is due to a lower exciton dissociation probability." Actually, it is believed that the higher dopant concentrations may also benefit the increase of recombination probability associated with the denser triplet generation, which seems not be the situation and should be further explained.
- (2) From Extended Data Figure 1d, it was observed that at 470 nm, the absorption intensity of CBP and T2T seems not to be zero. Also, for better comparison, it is believed that different excitation wavelength should be adopted for comparison.
- (3) As can be seen from Extended Data Figure 1, with dopant concentration increasing, the PL spectra of the films were gradually redshifted. The redshifted spectra indicate that the concentration of TADF materials significantly affect the energy levels, which may also lead to the different PL decay curves. And this possibility should be ruled out.
- (4) In Figure 1b, with dopant concentration increases, the long-tail gradually reduced. While in Extended Data Figure 3, the situation is different, which should be explained.
- (5) In Figure 3d, it seems that long tail was also observed for TBRb, which should be further explained.
- (6) As shown in Figure 2c and 2d, The TRPL tail intensity after applying the external electric field becomes appreciably weaker compared to the intensity when an electric field was not applied. However, it is noticed that despite the delay time of the voltage applied, the long-tail of the films are almost the same, which need more explanations for better understanding.
- (7) The author should give some comments about why choosing TPA-DCPP as the model. It is known that in the past years, some non-doped TADF materials have been developed (Takuma Yasuda et al, *Adv. Mater.* 2017, 29, 1604856), which showed no change in the long-tail with the increasing concentration. Such materials should be investigated for comparison. And they should give more results about 4CzIPN rather than only one PL decay curves.
- (8) The author should also explain why it is the triplet excitons that dissociate rather than the singlet ones.

Reviewer #3 (Remarks to the Author):

This manuscript investigates the mechanism for the observed concentration dependent decrease of fluorescence from the donor-acceptor molecule TPA-DCPP doped in a matrix of 4'-bis(carbazole-9-yl) biphenyl (CBP). The decrease in fluorescence yield for this highly polar TADF molecule is assigned to triplet state exciton dissociation (leading to charge carrier generation). An impressive range of time-resolved photoluminescence and electroluminescence measurements have been applied that confirm the presence of charge carriers and provide insight in to the role of external and internal electric fields on charge carrier behaviour. Excited state dissociation to form charges

in solid media is well known (e.g. blinking in single molecule fluorescence measurements is often assigned to fluorophore-matrix mediated charge separation and recombination) but the importance of this work rests on the insights into the concentration dependent behaviour and the general applicability to TADF molecules which are finding a variety of applications. The experimental work looks appropriately done and the conclusions are reasonable based on the results obtained. The research can be important to the field but in order to demonstrate the high impact of the work to readers, the authors need to address the following.

(i) The photoluminescence quantum yield for TPA-DCPP decreases substantially (0.6 to < 0.2) over the concentration range 0 - 100 wt% as shown in Figure 1a. The manuscript assigns a reason for this decrease as triplet exciton dissociation. However, it is not made clear if this is the only contribution to the decrease in emission observed. What percentage of this decrease is assigned to the dissociation/charge-generation mechanism? This needs to be explicitly quantified to demonstrate the importance of the process.

(ii) Do second order triplet processes play any role at higher concentrations of the fluorophore? These can lead to excited state dissociation and might be expected at the higher TPA-DCPP loadings.

(iii) Is there any evidence for a concentration dependent degradation of the fluorophores that could imply bimolecular processes play any role?

(iv) The authors should clarify the role of the CBP matrix. Why is this used rather than other matrices and what effect does changing the matrix make?

Reviewer #1 (Remarks to the Author):

In this manuscript, Yamanaka et al studied the long tail of fluorescence emissions and assigned its origination to the spontaneous exciton dissociation in the solid state. Though they pointed some interesting results, some statements are not so accurate. For materials with CT states, it has been widely recognized that charge recombination and dissociation exist simultaneously, which has been studied in photovoltaic systems. Therefore, for TADF materials, such processes would coexist and such behaviors can vary with different molecules and different dopant concentrations. Though the author demonstrated the dissociation process of TADF emitters, they did not provide more information about how to control it to enhance the performance of TADF emitters in devices. Therefore, I think major revisions are required:

[Reply]

First, we sincerely thank Reviewer #1 for taking the precious his/her time to evaluate our manuscript and provide very insightful comments and suggestions, which have greatly helped us further improve the quality of our manuscript. We agree with the reviewer's point that the charge recombination and dissociation event exist simultaneously in charge-transfer (CT)-type molecular systems. However, we would like to emphasize that the main claim of this study is the discovery that dissociated excitons diffuse spontaneously by 'concentration-dependence spontaneous orientation polarization' of the organic film even without any external electrical field. These phenomena have not been reported, and we believe that it can strongly contribute to understand the fundamental physics of organic semiconductor devices including not only OLEDs but also photovoltaic devices that use CT-type molecular systems. Further, as we mentioned in our manuscript, exciton dissociation during OLED operation is negligibly small. This is because the direction of the external electric field favors carrier recombination rather than exciton dissociation. The problem is that there is an additional exciton quenching process under optical excitation that does not exist under electrical excitation. This is the point that we would like to point out in this manuscript. Thus, we believe that Reviewer #1 can understand the importance of our findings. To emphasize this point, we added the sentence as follow:

Additional sentence (page 9 line 14):

'Additionally, the long-lived decay disappears completely in transient electroluminescence profiles (**Extended Data Fig. 3e**) of OLEDs because the applied voltage promotes carrier recombination rather than charge separation. **This means that there is an additional exciton quenching process under optical excitation that does not exist under electrical excitation condition.**'

Further, we have carefully revised the original manuscript based on the reviewer's comments as follows.

(1) In the manuscript, the author found that “the TRPL decay diminishes with an increase of excitation power density. This can be understood as a shortening of the diffusion length of the carriers because of an increase of recombination probability associated with the denser triplet generation” and “the intensity of the emission spike in a 20-wt%-doped non-injecting device is much weaker than that of a 50-wt%-doped one (Extended Data Fig. 3d), which is due to a lower exciton dissociation probability.” Actually, it is believed that the higher dopant concentrations may also benefit the increase of recombination probability associated with the denser triplet generation, which seems not be the situation and should be further explained.

[Reply]

When a 20-wt%-doped sample and a 50-wt%-doped sample are compared, their exciton dissociation probability and carrier diffusion probability are different, i.e., these probabilities in the 50-wt%-doped sample are higher than those in the 20-wt%-doped sample. Therefore, even if the exciton recombination probability is also large in the 50-wt%-doped sample, the total amount of the separated charges can be expected to be large. On the other hand, when the dependence of the excitation intensity is measured in the same sample, a high exciton density results in a short charge diffusion length due to an increase of the recombination probability. Therefore, the two sentences are consistent with each other.

(2) Form Extended Data Figure 1d, it was observed that at 470 nm, the absorption intensity of CBP and T2T seems not to be zero. Also, for better comparison, it is believed that different excitation wavelength should be adopted for comparison.

[Reply]

As shown in the attached figure, even if we change the excitation wavelength at 340 nm or 470 nm, the 50-wt%-doped sample showed the distinct long-tail emission. The difference of the long-tail emission intensity is due to the difference of the absorbance at each excitation wavelength, i.e., the difference of exciton density as discussed above. The increase in the background of the absorption spectrum at the long wavelengths was caused by the changes of the reflectance and the interference effect between a reference sample and a sample. Thus, we changed the sentence to avoid misunderstanding.

Attached Figure 1: Dependence of the TRPL decay profile on the excitation light wavelength for a 50-wt%-TPA-DCPP:CBP blend.

Original sentence:

‘Additionally, the T2T and CBP layers contain no absorption bands at the excitation wavelength (470 nm), meaning that only TPA-DCPP is directly excited (**Extended Data Fig. 1d**).’

Revised sentence (page 5 line 5):

‘Additionally, the T2T and CBP layers contain **almost no absorbance** at the excitation wavelength (470 nm), meaning that TPA-DCPP is mainly excited (**Extended Data Fig. 1d**).’

(3) As can be seen from Extended Data Figure 1, with dopant concentration increasing, the PL spectra of the films were gradually redshifted. The redshifted spectra indicate that the concentration of TADF materials significantly affect the energy levels, which may also lead to the different PL decay curves. And this possibility should be ruled out.

[Reply]

We appreciate the insightful comment by the reviewer. The observation of the redshifted spectrum with increasing the doping concentration is not conflicting to our proposed mechanism. Generally, the energy stabilization of the charge-transfer (CT) state can be explained by the change of the polarity of the environment such as solvent, *i.e.*, solvation. The redshift of the emission spectrum with the increasing doping concentration in the host-guest solid-state film means the change of the polarity of the matrix. Since the increase of the doping concentration of TPA-DCPP induced a large SOP as we revealed in the manuscript, this trend of the emission spectrum change is natural. These behaviors have been reported as ‘self-solid-state solvation’ (Ref: H.S. Kim et al., J. Phys. Chem. C, 2017, 121, 13986), while the effect of the dissociation of excitons by the exciton quenching has not been discussed. Although it is difficult at present to quantitatively separate the effects of exciton dissociation by SOP or energy stabilization by SOP on the PL decay curve, it is evident from our experimental evidence that exciton dissociation (and charge diffusion) by SOP occurs. We added the sentence to discuss this point.

Additional sentence (page 7 line 14):

‘Based on the DCM results, we can explain the driving force for carrier generation in the solid films. Since TPA-DCPP has a large permanent dipole moment (13.05 D), the threshold-voltage shift observed in the DCM (**Fig. 3a**) can be attributed to the formation of a concentration-dependent potential slope across the EML by spontaneous orientation polarization (SOP)^{17,18}. **Since the concentration-dependent potential slope means a change of polarity of the EML with increasing the doping concentration, the PL spectrum of the highly doped codeposited film shows a redshifted spectrum as shown in Extended Data Fig. 1b. In other words, the observation of the redshifted spectrum with increasing the doping concentration is not conflicting with our proposed mechanism.**’

(4) In Figure 1b, with dopant concentration increases, the long-tail gradually reduced. While in Extended Data Figure 3, the situation is different, which should be explained.

[Reply]

We thank the Reviewer for pointing out the difference of the TRPL behavior and apologize for the confusing statement in the original manuscript. The time scale of Fig. 1b is much shorter than that of Extended Data Fig. 3d, and the excitation light pulse width used in Fig. 1b is much shorter (10 μ s) than that used in Extended Data Fig. 3d (500 μ s). The disappearance of the delayed component with increasing of the doping concentration (Fig. 1b) means the quenching of the triplet excitons. As we revealed, the spontaneous exciton dissociation and the diffusion of the carriers are the origin of the ‘long-tail’ emission when the film was excited with the long-excitation pulse width. Thus, the much intense long-tail component in the highly doped sample should be observed (Extended Data Fig. 3d). To prevent any confusion, we revised the sentence as follow.

Original sentence:

‘These observations indicate that, following long-pulse excitation, a long-lived emission component that is different from “normal” triplet exciton decay occurs in the blend film even after the excitation light is turned off.’

Revised sentence (page 4 line 8):

‘These observations indicate that, following long-pulse excitation, a long-lived emission component that is different from “normal” triplet exciton decay, *i.e.*, TADF, occurs in the blend film even after the excitation light is turned off **when the film was excited by the long-excitation pulse width.**’

(5) In Figure 3d, it seems that long tail was also observed for TBRb, which should be further explained.

[Reply]

We apologize for lacking of the detailed discussion regarding the ‘strange’ tail that observed in the TBRb:CBP co-deposited film (Fig. 3d) and there is misdescription. In the experiments, the non-injection type devices based on these blend films as EML are used. In this measurement, since ultraviolet light (340 nm) was used as the excitation light, the ITO electrode slightly absorbed the excitation light and emitted light. In fact, the TRPL of the ITO substrate traces the TRPL curves of the TBRb sample (please see the attached figure). Further, since the device has a pair of electrodes (ITO and Al), an internal electric field is generated in a short circuit condition. This internal electric field may have a possibility to induce a small amount of exciton dissociation as shown in Fig. 2b. Thus, we revised the figure caption. However, note that the intensity of the long-tail emission component observed in the TBRb-based device is extremely small compared with that of the 4CzIPN-based device, indicating that the exciton dissociation by SOP hardly occurs in the TBRb-based device when the voltage offset set to 0 V. In fact, the EL spike by external voltage pulse (voltage condition: from 0 V to 5 V) in the device is negligibly small (please see the revised Fig. 3d). On the other hand, when negative voltage offset (−10 V) was applied to the device during measurement, the intense EL was clearly observed by the external voltage pulse (voltage condition: −10 V to 5 V) after the photoexcitation turn-off even in the TBRb-based device. This result indicates that the excitons of TBRb dissociate by the ‘external’ electrical field. In the case of the 4CzIPN-based device, however, the EL spike was clearly observed even when the voltage offset was set to 0 V, indicating that the excitons dissociate and diffuse even at this condition. These experimental results strongly support our main claim: ‘spontaneous exciton dissociation and carrier diffusion by SOP in the polar film’. We added these experimental results to Fig. 3d and added the sentence to support our conclusions.

Attached figure: TRPL profile of the ITO substrate and the TBRb-based sample.

Additional sentence (page 8 line 23):

‘Further, even when the voltage offset is set to 0 V, the non-injection-type devices based on 4CzIPN:CBP blends show clear EL spikes when an external voltage pulse was applied. This result indicates that the excitons of 4CzIPN are spontaneously dissociated and diffused by the SOP. Also, of course, when the voltage offset was set to −10 V on the TBRb-based device, a

clear response of the EL spike due to the external voltage pulse was observed. This result shows that the excitons of TBRb dissociate by the 'external' electric field. However, when the voltage offset was set to 0 V, no response was observed because of lacking of SOP.'

(6) As shown in Figure 2c and 2d, The TRPL tail intensity after applying the external electric field becomes appreciably weaker compared to the intensity when an electric field was not applied. However, it is noticed that despite the delay time of the voltage applied, the long-tail of the films are almost the same, which need more explanations for better understanding.

[Reply]

When a voltage pulse is applied, the diffused charges can be recombined. Therefore, the long-tail emission intensity decreased after the voltage pulse was applied. The experiment in Figure 2c showed that the EL was observed even when a voltage was applied at an arbitrary timing. In this experiment, we used independent voltage pulses for each timing and multiple measurements were performed. Note that it is not a multiple voltage pulse. To prevent any misleading for general readers, we revised the figure caption of Fig. 2c.

Original sentence:

'c, Dependence of the external-electric-field-modulated TRPL profiles on external-voltage-pulse delay in the 50 wt%-TPA-DCPP:CBP device. The applied voltage and the width were fixed to 5 V and 50 μ s, respectively.'

Revised sentence (figure caption in Fig.2):

'c, Dependence of the external-electric-field-modulated TRPL profiles on external-voltage-pulse delay in the 50 wt%-TPA-DCPP:CBP device. The applied voltage and the width were fixed to 5 V and 50 μ s, respectively. Note that it is not a multiple voltage pulses. This experiment used independent voltage pulses for each timing, and multiple measurements were performed.'

(7) The author should give some comments about why choosing TPA-DCPP as the model. It is known that in the past years, some non-doped TADF materials have been developed (Takuma Yasuda et al, Adv. Mater. 2017, 29, 1604856), which showed no change in the long-tail with the increasing concentration. Such materials should be investigated for comparison. And they should give more results about 4CzIPN rather than only one PL decay curves.

[Reply]

To demonstrate a proof-of-concept of our proposal, we selected TPA-DCPP as a model compound because it possesses an impressive large permanent dipole moment. In addition to this reason, to exclude any effect of the permanent dipole moment of the host matrix on the

observations, we should choose non-polar host materials, *i.e.*, CBP. The triplet energy level of CBP lies around 2.55 eV. Thus, it was necessary to use red or green emissive materials that utilize triplet exciton instead of blue ones. We revised the sentence to explain this point.

Original sentence:

‘To find evidence of spontaneous exciton dissociation in organic fluorophore alone in a solid matrix, we first evaluated the fundamental photoluminescence (PL) properties of a model fluorophore, namely, TPA-DCPP¹³ (Fig. 1a; dipole moment = 13.05 D).’

Revised sentence (page 3 line 11):

‘To find evidence of spontaneous exciton dissociation in organic fluorophore alone in a solid matrix, we first evaluated the fundamental photoluminescence (PL) properties of a model fluorophore, namely, TPA-DCPP¹³ (Fig. 1a) because it possesses a large permanent dipole moment (dipole moment = 13.05 D).’

Original sentence:

‘Based on the DCM results, we can explain the driving force for carrier generation in the solid films. Since TPA-DCPP has a large permanent dipole moment (13.05 D), the threshold-voltage shift observed in the DCM (Fig. 3a) can be attributed to the formation of a concentration-dependent potential slope across the EML by spontaneous orientation polarization (SOP)^{17,18}.’

Revised sentence (page 7 line 10):

‘Based on the DCM results, we can explain the driving force for carrier generation in the solid films. Since a permanent dipole moment of CBP is close to zero and TPA-DCPP has a large permanent dipole moment (13.05 D), the threshold-voltage shift observed in the DCM (Fig. 3a) can be attributed to the formation of a concentration-dependent potential slope across the EML by spontaneous orientation polarization (SOP) of TPA-DCPP molecules^{17,18}.’

Next, we mention about the reviewer’s suggestions regarding the recently developed highly emissive TADF material that shows high PLQY even in a neat film (T. Yasuda et al, Adv. Mater. 2017, 29, 1604856). As mentioned above, the TADF materials exhibiting blue or sky-blue emission are not suitable for our analysis because it is difficult to select a non-polar host material to prevent an back-energy transfer process. However, based on the physics found in this study, we would like to continue to study exciton dynamics in molecular systems exhibiting high or low PLQY with TADF ability in highly-doped films. In particular, we are interested in an exciplex system (50 mol% donor : 50 mol% acceptor). We have already found that in molecular systems showing high PLQY, exciton dissociation hardly occurs, and have also clarified that such

molecular systems do not have a large SOP. We will report these experimental data as a separated paper as soon as possible. Further, we added some additional experimental data of the 4CzIPN-based device that reinforce our proposed mechanism in Fig. 3d.

(8) The author should also explain why it is the triplet excitons that dissociate rather than the singlet ones.

[Reply]

We really thank the reviewer for pointing out the role of spin-state for the exciton dissociation event. We carefully reconsidered this point. The triplet excitons of TADF materials are considered to dissociate because of (1) long excited state lifetime, and (2) the small energy splitting between singlet CT and triplet CT energy levels. Moreover, the disappearance of the delayed fluorescence component with increasing doping concentration observed in the TPA-DCPP:CBP solid-state films strongly suggests the quenching of the triplet excitons. However, after careful consideration, although it does not deny the main conclusion of this study, ‘spontaneous exciton dissociation by SOP’, at this stage it is rather hard to determine which spin state, singlet and triplet, is responsible mainly for the exciton dissociation. We have revised the manuscript to account for these considerations, and the assertive descriptions were deleted. In addition, we corrected the reference #12.

Additional sentence (page 8 line 13):

‘Here, we discuss the role of spin states in the spontaneous exciton dissociation by the SOP. The triplet excitons of polar molecules, *i.e.*, CT-type molecule exhibiting TADF, are considered to dissociate because of (1) the long exciton lifetime, and (2) the small energy splitting between singlet CT and triplet CT energy levels. Further, the disappearance of the delayed fluorescence component with increasing the doping concentration suggests the quenching of the triplet excitons. However, it cannot be determined which spin state, singlet and triplet, is responsible for the exciton dissociation at this stage. While we need to study it in more detail, it is evident that excitons are spontaneously dissociated by SOP.’

Reviewer #3 (Remarks to the Author):

This manuscript investigates the mechanism for the observed concentration dependent decrease of fluorescence from the donor-acceptor molecule TPA-DCPP doped in a matrix of 4'-bis(carbazole-9-yl) biphenyl (CBP). The decrease in fluorescence yield for this highly polar TADF molecule is assigned to triplet state exciton dissociation (leading to charge carrier generation). An impressive range of time-resolved photoluminescence and electroluminescence measurements have been applied that confirm the presence of charge carriers and provide insight in to the role of external and internal electric fields on charge carrier behaviour. Excited state dissociation to form charges in solid media is well known (e.g. blinking in single molecule fluorescence measurements is often assigned to fluorophore-matrix mediated charge separation and recombination) but the importance of this work rests on the insights into the concentration dependent behaviour and the general applicability to TADF molecules which are finding a variety of applications. The experimental work looks appropriately done and the conclusions are reasonable based on the results obtained. The research can be important to the field but in order to demonstrate the high impact of the work to readers, the authors need to address the following.

[Reply]

We thank Reviewer #3 for taking efforts in evaluating our manuscript and the positive assessment. We also sincerely thank the reviewer for his/her insightful comments and suggestions, which have strongly helped us further improve our manuscript. Based on the reviewer's comments, we have carefully revised the original manuscript as follows.

(i) The photoluminescence quantum yield for TPA-DCPP decreases substantially (0.6 to < 0.2) over the concentration range 0 - 100 wt% as shown in Figure 1a. The manuscript assigns a reason for this decrease as triplet exciton dissociation. However, it is not made clear if this is the only contribution to the decrease in emission observed. What percentage of this decrease is assigned to the dissociation/charge-generation mechanism? This needs to be explicitly quantified to demonstrate the importance of the process.

[Reply]

We appreciate the comments and apologize for not inserting more detailed discussion regarding the concentration quenching for PLQY in our original manuscript. We fully agree with the reviewer's point. Several competing exciton quenching processes such as dipole-dipole interaction and nonradiative decay from a triplet to a ground state should be considered to explain the concentration quenching behavior. Thus, although the exciton dissociation event by spontaneous orientation polarization (SOP) is one of the quenching processes, it may not be the main channel for the concentration quenching. Regarding the yield of the dissociation event, as we mentioned in page 8, the yield of the long-lived emission to the total emission is calculated to

be approximate ~1%. However, the dissociation yield of the excitons should be much higher than this yield. We revised the sentence to emphasize this point and prevent any misreading for general readers.

Original sentence:

‘To find evidence of spontaneous exciton dissociation in organic fluorophore alone in a solid matrix, we first evaluated the fundamental photoluminescence (PL) properties of a model fluorophore, namely, TPA-DCPP¹³ (Fig. 1a; dipole moment = 13.05 D). The PL quantum yield (PLQY) of TPA-DCPP doped in a CBP host matrix at different concentrations (the absorption and emission spectra are shown in Extended Data Fig. 1a,b) decreased with an increase of the doping concentration as shown in Fig. 1a.’

Revised sentence (page 3 line 14):

‘To find evidence of spontaneous exciton dissociation in organic fluorophore alone in a solid matrix, we first evaluated the fundamental photoluminescence (PL) properties of a model fluorophore, namely, TPA-DCPP¹³ (Fig. 1a; dipole moment = 13.05 D). The PL quantum yield (PLQY) of TPA-DCPP doped in a CBP host matrix at different concentrations (the absorption and emission spectra are shown in Extended Data Fig. 1a,b) decreased with an increase of the doping concentration as shown in Fig. 1a, **indicating the concentration quenching processes such as the deactivation of exciton via dipole-dipole interaction and the internal conversion process coexist in the highly doped films.** Here note that the exciton quenching due to a back energy transfer from T₁ of TPA-DCPP (2.25 eV) to T₁ of CBP (2.55 eV) is negligible because triplet energy level of CBP matrix higher than that of TPA-DCPP.’

Original sentence:

‘We roughly calculate the yield of long-lived emission to total emission is ~1%, so the dissociation yield should be much higher than this.’

Revised sentence (page 8 line 12):

‘We roughly calculate the yield of long-lived emission to total emission is ~1%, so the dissociation yield should be much higher than this. **Compared with the total exciton quenching yield, the dissociation yield is actually quite small, but the fact that the process exists cannot be ignored.**’

(ii) Do second order triplet processes play any role at higher concentrations of the fluorophore? These can lead to excited state dissociation and might be expected at the higher TPA-DCPP loadings.

(iii) Is there any evidence for a concentration dependent degradation of the fluorophores that could imply bimolecular processes play any role?

[Reply]

We fully agree with the point that we should exclude the effect of the bimolecular process such as triplet-triplet annihilation events on our observations. We carefully check the dependence of the excitation power on the emission intensity in the 50wt%-TPA-DCPP doped CBP film and confirmed the almost linear response of the emission intensity with the excitation power. This result clearly indicates that the bimolecular process does not strongly contribute to our observations at the excitation power range used in this study. We added the sentence to emphasize this point.

Additional sentence (page 4 line 9):

‘Further, since the emission intensity in a 50-wt%-TPA-DCPP:CBP film shows the almost linear response for the excitation light power (Extended Data Fig. 1d), the effect of the bimolecular exciton annihilation process can be excluded.’

Extended Data Fig.1e: Dependence of the PL intensity on the excitation light power for a 50-wt%-TPA-DCPP:CBP blend.

(iv) The authors should clarify the role of the CBP matrix. Why is this used rather than other matrices and what effect does changing the matrix make?

[Reply]

We really thank the Reviewer for pointing out the role of the host matrix. Since the permanent dipole moment of CBP molecule is close to zero (Ref: Y. Noguchi et al., J. Appl. Phys., 2012, 111, 114508), we can exclude the effect of spontaneous orientation polarization (SOP) that induced by the CBP molecules. Thus, we conclude that the observed SOP in the TPA-DCPP doped CBP film is due to a large permanent dipole moment of TPA-DCPP molecules (13.05 D). In addition to the reason, the triplet energy level of CBP matrix (2.55 eV) is higher than that of TPA-DCPP (2.25 eV). Therefore, we can also exclude the effect of the triplet exciton quenching by back-energy transfer from TPA-DCPP to CBP host molecule. We revised the sentences to explain the reason that we chose CBP as the host matrix.

Original sentence:

‘To find evidence of spontaneous exciton dissociation in organic fluorophore alone in a solid matrix, we first evaluated the fundamental photoluminescence (PL) properties of a model fluorophore, namely, TPA-DCPP¹³ (**Fig. 1a**; dipole moment = 13.05 D). The PL quantum yield (PLQY) of TPA-DCPP doped in a CBP host matrix at different concentrations (the absorption and emission spectra are shown in Extended Data Fig. 1a,b) decreased with an increase of the doping concentration as shown in Fig. 1a.’

Revised sentence (page 3 line 16):

‘To find evidence of spontaneous exciton dissociation in organic fluorophore alone in a solid matrix, we first evaluated the fundamental photoluminescence (PL) properties of a model fluorophore, namely, TPA-DCPP¹³ (**Fig. 1a**; dipole moment = 13.05 D). The PL quantum yield (PLQY) of TPA-DCPP doped in a CBP host matrix at different concentrations (the absorption and emission spectra are shown in Extended Data Fig. 1a,b) decreased with an increase of the doping concentration as shown in Fig. 1a, indicating the concentration quenching processes such as the deactivation of exciton *via* dipole-dipole interaction and the internal conversion process coexist in the highly doped films. **Here note that the exciton quenching due to a back energy transfer from T₁ of TPA-DCPP (2.25 eV) to T₁ of CBP (2.55 eV) is negligible because triplet energy level of CBP matrix is higher than that of TPA-DCPP.**’

Original sentence:

‘Based on the DCM results, we can explain the driving force for carrier generation in the solid films. Since TPA-DCPP has a large permanent dipole moment (13.05 D), the threshold-voltage shift observed in the DCM (**Fig. 3a**) can be attributed to the formation of a concentration-dependent potential slope across the EML by spontaneous orientation polarization (SOP)^{17,18}.’

Revised sentence (page 7 line 10):

‘Based on the DCM results, we can explain the driving force for carrier generation in the solid films. Since **a permanent dipole moment of CBP is close to zero and** TPA-DCPP has a large permanent dipole moment (13.05 D), the threshold-voltage shift observed in the DCM (**Fig. 3a**) can be attributed to the formation of a concentration-dependent potential slope across the EML by spontaneous orientation polarization (SOP) **of TPA-DCPP molecules**^{17,18}.’

REVIEWERS' COMMENTS:

Reviewer #1 (Remarks to the Author):

The revised manuscript is acceptable as is.

Reviewer #2 (Remarks to the Author):

According to authors' response and text, I think the manuscript is now suitable for publication in the journal.

Reviewer #3 (Remarks to the Author):

In this revised version the authors have satisfactorily addressed the issues I raised in my initial review with their additional comments. While the contribution that the triplet exciton dissociation makes to the concentration quenching has not been precisely quantified, the authors do now acknowledge in the text that the contribution is small which will be important information for readers. In my view the manuscript is now acceptable for publication.